# Impact of Different Temperatures on Activity of the Pest *Monolepta hieroglyphica* Motschulsky (Coleoptera: Chrysomelidae)

**DOI:** 10.3390/insects16020222

**Published:** 2025-02-18

**Authors:** Rongrong Shi, Jianyu Hao, Yue Zhang, Qinglei Wang, Chunqin Liu, Qing Yang

**Affiliations:** 1Hebei Key Laboratory of Soil Entomology, CangZhou Academy of Agriculture and Forestry Sciences, Cangzhou 061001, China; shirongrongabc@163.com (R.S.); hjy06231818@163.com (J.H.); zhyue1028@163.com (Y.Z.); wqlei02@163.com (Q.W.); 2Doctoral Work Laboratory, Department of Agricultural and Animal Husbandry Engineering, Cangzhou Technical College, Cangzhou 061001, China

**Keywords:** temperature stress, climate adaptability, antioxidant response

## Abstract

**Simple Summary:**

As poikilotherms, insects are sensitive to ambient environmental conditions; therefore, it is important to gauge how heat stress affects their survival and fitness. The leaf beetle *Monolepta hieroglyphica* (Motschulsky) is a key pest within farmlands in China. This study investigated the effects of different temperatures (i.e., 25, 28, 31, and 34 °C) on the survival, reproduction, feeding capacity, and antioxidant capacity of adult leaf beetles. Laboratory assays showed that elevated temperatures (i.e., 31 and 34 °C) had a negative effect on the survival and reproduction of *M. hieroglyphica*. As temperatures increased (25 °C to 34 °C), the feeding capacity of *M. hieroglyphica* decreased significantly. Lastly, superoxide dismutase (SOD), catalase (CAT), glutathione-s-transferase (GST), and peroxidases (POD) in leaf beetles were significantly affected by ambient temperature. By assessing the thermal biology of leaf beetles, these laboratory assays can provide a theoretical basis for the development of integrated pest management (IPM) programs for *M. hieroglyphica*.

**Abstract:**

*Monolepta hieroglyphica* (Motschulsky) (Coleoptera: Chrysomelidae) is widely distributed in China and is polyphorous, being a major pest to cash crops, such as corn, cotton, and millet. Given the increasing severity of the greenhouse effect in recent years, we aimed to investigate the adaptability of *M. hieroglyphica* adults to varying temperatures. In this study, we assessed the survival, longevity, fecundity, feeding capacity, and antioxidant capacity of leaf beetle adults under laboratory conditions at 25–34 °C. Elevated temperatures (i.e., 31 and 34 °C) had (negative) impacts on *M. hieroglyphica* adults’ survival and reproduction. Similarly, the temperature negatively affected the feeding capacity of *M. hieroglyphica* adults, with the impact becoming more pronounced as the temperature increased. Under the same treatment time, the SOD and CAT activity levels increased with the increase in treatment temperature. The GST activity levels showed a decreasing trend. The POD activity showed a biphasic response to increasing temperatures, first decreasing and then increasing. The above indicates that different antioxidant enzymes of *M. hieroglyphica* adults have different levels of sensitivity to high temperatures. In the laboratory, our work analyzes the response of *M. hieroglyphica* adults to temperature from ecological and physiological research perspectives and provides strategies for strengthening its subsequent integrated pest management (IPM) under conditions of global warming or extreme weather events.

## 1. Introduction

The widespread adoption of transgenic insect-resistant crops has effectively controlled target lepidopteran pests. However, the threat posed by coleopteran insects may escalate from a secondary to a primary concern, warranting our attention. Among these, *Monolepta hieroglyphica* Motschulsky (Coleoptera: Phyllotaxae) is a polyphagous pest that is widely distributed in China and globally [1,2,3]. This species is capable of shifting its primary threat among various cash crops and weeds, such as corn, cotton, purslane, etc [4,5]. In recent years, *M. hieroglyphica* has emerged as a significant pest in China’s inland crops, including corn, cotton, and soybeans [6]. This pest can cause plant deformities, disrupt crop pollination, and ultimately result in substantial reductions in crop yields [7,8,9]. Furthermore, the geographical range of *M. hieroglyphica* has expanded, starting from Heilongjiang and Inner Mongolia in the north, extending to Taiwan, Guangdong, Guangxi, and Yunnan in the south, reaching the border in the east, and extending to Ningxia, Gansu, Sichuan, and Yunnan in the west. Its rate of spread has accelerated [6], marking this insect as a key concern.

*M. hieroglyphica*, as poikilotherms, are highly sensitive to temperature changes in the external environment. Numerous studies have found that temperature has a significant impact on the growth, development, reproductive ability, and physiological metabolism of insects [10,11,12]. Laboratory studies have shown how elevated temperatures affect growth, development, and survival during the immature and adult stages of various leaf beetle species [13,14,15]. The effects are species-specific, depend on the range of experimental temperatures, and similarly involve other life history parameters such as fecundity [16,17]. Another key (temperature-dependent) variable is the feeding capacity of leaf beetles [18]. Laboratory assays have demonstrated how temperature modulates the feeding capacity of various leaf beetle species, development stages, and host plants, e.g., for *M. hieroglyphica* adults on *Brassica pekinensis* (Lour.) and *Zea mays* (L.) [19], *Colposcelis signata* (Motschulsky) adults on *Glycine max* (Linn.) [20], or *Paridea angulicollis* (Motschulsky) larvae and adults on *Gynostemma pentaphyllum* (Thunb.) [21]. Therefore, in order to predict, forecast, prevent, and control leaf beetles in high-temperature production environments or under climate change scenarios, it is necessary to better understand these temperature-related impacts.

The induction of reactive oxygen species (ROS) in insects under high-temperature stress represents one of the most evident physiological changes; excessive levels of ROS can lead to oxidative toxicity in these organisms [22,23]. In order to effectively respond to changes in the environmental temperature, insects have formed efficient antioxidant systems during long-term evolution [24], in which antioxidant enzymes are key components in the regulation of intracellular ROS balance [25,26,27]. They mainly include catalase (CAT) and peroxidase (POD) that decompose high-concentration (H_2_O_2_), superoxide dismutase (SOD), which scavenges high-concentration superoxide anion radicals, and glutathione-S-transferase (GST) that metabolizes lipid peroxides, which can help insects to remove excess ROS from their bodies [28,29,30], which protects insects from oxidative damage.

This experiment investigates the effects of varying temperatures (25, 28, 31, and 34 °C) on the survival rate, longevity, fecundity, feeding capacity, and antioxidant capacity of *M. hieroglyphica*. Our specific objectives are (1) to compare the survival rates, longevity, and fecundity of adult leaf beetles at different temperatures; (2) to evaluate the feeding capacity of these beetles across the specified temperatures; and (3) to analyze the trends in antioxidant enzyme activity in these beetles at varying temperatures. The data obtained will provide a theoretical basis for the development of integrated pest management (IPM) programs for this leaf beetle.

## 2. Materials and Methods

### 2.1. Insect Rearing

The *M. hieroglyphica* species were collected from corn fields (pesticide-free) at Xinzhou (38.26 °N, 112.39 °E) (Xinzhou, Shanxi Province) in 2023. The key characteristics used to determine the species of *M. hieroglyphica* were, for example, a near-circular pale spot found on each elytra base, etc [31,32]. Next, the field-caught individuals were transferred to the Institute of Plant Protection of the Cangzhou Academy of Agricultural and Forestry Sciences (CAAS; 38.16 °N, 116.48 °E) in Cangzhou, Hebei Province. The *M. hieroglyphica* species were kept under laboratory conditions. The *M. hieroglyphica* were reared on *Zea mays* L. (corn: five-leaf stage of variety Longsheng 802, Jinzhong Longsheng Seed Co., Ltd., Xinzhou China) in screened cages (30 cm × 30 cm × 30 cm) within a controlled climate chamber (RXZ500D, Ningbo Jiangnan Instrument Factory, Ningbo, China) and held at 25 ± 1 °C, 70 ± 5% RH, and 16:8 h (light/dark) photoperiod. The *Zea mays* plants were replaced every two days to ensure the stability of the *M. hieroglyphica* population. The *F*_1_ progeny was used for this experiment.

### 2.2. Temperature Treatments

All subsequent experimental assays were performed in the lab and carried out in climatic chambers with temperatures of 25 °C, 28 °C, 31 °C, and 34 °C, which mirror the prevailing temperatures during the corn and other crops’ growing seasons under the background of the greenhouse effect in north China. Combined with the pre-experimental results, the above four temperatures (25 °C, 28 °C, 31 °C, and 34 °C) were determined. All climatic chambers were monitored using HOBO to ensure constant temperatures were maintained.

### 2.3. Adult Survival and Reproduction

*F*_1_ adults of *M. hieroglyphica* < 12 h of age were removed from the rearing colony, placed within screened cages (30 cm × 30 cm × 30 cm), and fed corn ad libitum. Leaf beetles were kept at 25 °C, and adults were allowed to freely oviposit. Each day, the egg boxes (12.5 cm length, 8 cm width, containing a water-moistened cotton pad and gauze) were transferred to a new Petri dish (7.5 cm diam, 1.5 cm high) to await incubation, and the adults were left in place until they died. The newly emerged larvae were then transferred to new containers and fed freely with corn until pupation. Upon pupal emergence, 50 pairs of F_2_ adults (<12 h of age) were randomly selected and placed into climatic chambers at 25, 28, 31, or 34 °C, 70 ± 5% RH, and 16:8 h (L/D) photoperiod, respectively. Each 50 pairs of leaf beetles were placed into screened cages (30 cm × 30 cm × 30 cm) containing an egg box and corn (five-leaf stage of corn plant). On a daily basis, leaf beetle survival and the number of deposited eggs were recorded, with the observations continuing until all the adults died. There were three replicates for each temperature condition (i.e., treatment) and 50 pairs of leaf beetles per replicate.

### 2.4. Adult Feeding Capacity

As described by Kaufmann [33], we assessed the feeding capacity of *M. hieroglyphic* at the four experimental temperatures and above climatic conditions (70 ± 5% RH, 14:10 (L/D) photoperiod). At the onset of the experiment, the *M. hieroglyphic* adult females (<12 h of age) were individually starved for 24 h. Next, five leaf beetles were transferred within a screened cage (15 cm × 15 cm × 15 cm) containing a corn leaf (pre-weighed). The screened cages without leaf beetle adults were used as a control. At each experimental temperature, a total of twenty-five leaf beetle adults (i.e., replicates) were individually exposed to a given quantity of food items for 24 h. Next, we recorded the quantity of corn consumed by each of the five leaf beetle adults.

Following the below formula [33]:Correct feeding amount = *W* − [*L* + (*aW* + *bL*)/2]

In which *W* is the weight of the test food at the beginning of the experiment, *L* is the weight of the remaining food at the end of the experiment, *a* = (the initial weight of the control group food—the final weight of the control group food)/the initial weight of the control group food, *b* = (the initial weight of the control group food—the final weight of the control group food)/the final weight of the control group food [33].

### 2.5. Antioxidant Responses

The adults of *M. hieroglyphica* < 12 h of age were placed within screened cages (55 cm × 35 cm × 50 cm) and subjected to 24 h, 48 h, and 72 h under the experimental temperatures, i.e., 25, 28, 31, and 34 °C. Next, live, healthy adults (male and female) were chosen, quickly immersed in liquid nitrogen, and then they were stored in an −80 °C refrigerator until further laboratory testing. Next, the frozen individuals were placed in phosphate-buffered saline (PBS, PH 7.4) at a ratio of 1 mL of PBS for each 0.1 g of body tissue. The samples were placed in a cold mortar, homogenized with liquid nitrogen, and crude extracts were centrifuged at 4 °C and 10,000× *g* for 10 min. The supernatant was then centrifuged under the above conditions to determine the antioxidant capacity. There were three replicates for each treatment and one pair of leaf beetles in each replicate.

The activity levels of four antioxidant enzymes (i.e., SOD, CAT, GST, and POD) were determined using commercial assay kits (Jianglaibio Co., Ltd., Shanghai, China) following the manufacturer’s instructions. Absorbance was recorded using a light-absorbing enzyme marker (BioTek 800 ™ TS, BioTek Co., Ltd., Winooski, VT, USA), with the activities of SOD, CAT, GST, and POD being detected at 450 nm.

### 2.6. Data Analysis

A one-way analysis of variance was used to analyze the effect of the temperatures on the adult longevity, fecundity, feeding capacity, and antioxidants of leaf beetles. Meanwhile, a two-way analysis of variance was used to analyze the effects of the temperature and time on the adult antioxidants of leaf beetles (*p* < 0.05). Tukey’s test was used to determine the differences between different temperatures for the leaf beetle (*p* < 0.05). Survival curves of the leaf beetle in between different temperatures were analyzed by the Kaplan–Meier log-rank test. All statistical analyses were conducted using SPSS 25.0 software and Microsoft Excel 2010, while the charts were generated using SigmaPlot 12.5 and OriginPro 9.0.

## 3. Results

### 3.1. Adult Survival and Reproduction

The survival of *M. hieroglyphica* adults was affected by the temperature (log-rank test: *χ*^2^ = 699.81, *df* = 2, *p* < 0.001, Figure 1A). With the increase in temperature (25 °C to 28 °C to 31 °C to 34 °C), the survival rate of the *M. hieroglyphica* adults decreased significantly. At 34 °C, the survival rate of the *M. hieroglyphica* adults decreased most significantly (Tukey’s test: *F*_3,8_ = 6040.50, *p* < 0.001, Figure 1A). There was a significant difference in the survival rate of *M. hieroglyphica* adults on the first day of treatment compared to other temperatures. Therefore, a high temperature (34 °C) has a significant negative effect on the *M. hieroglyphica* species.

Similarly, the temperature had an effect on the longevity of *M. hieroglyphica* adults (Tukey’s test: *F*_3, 8_ = 3018.15, *p* < 0.001, Figure 1B). For *M. hieroglyphica*, further temperature increases also lowered their longevity, declining from 22.35 d (25 °C) to 18.79 d (28 °C) to 16.34 d (31 °C) to 11.53 d at 34 °C, respectively (Figure 1B).

An elevated temperature equally affected the age-specific and total fecundity of *M. hieroglyphica* adults (Figure 2A). Oviposition rates of *M. hieroglyphica* declined from 360.70 (25 °C) to 199.43 (28 °C) to 10.99 (31 °C) to 0 (34 °C), respectively (Tukey’s test: *F*_3, 8_ = 35,475.16, *p* < 0.001, Figure 2B). Therefore, a high temperature limits the reproduction of *M. hieroglyphica*.

### 3.2. Adult Feeding Capacity

The results of the *M. hieroglyphica* adult daily food consumption are shown in Figure 3. Under 25–34 °C, the daily food consumption of *M. hieroglyphica* adults decreased with the increase in temperature (Tukey’s test: *F*_3, 8_ = 1007.83, *p* < 0.001, Figure 3). Compared with 25 °C, at 28 °C, there was no significant change in the daily food consumption of corn leaves by the single-headed *M. hieroglyphica* adult. However, when the temperature increased to 31 °C and 34 °C, the daily food consumption decreased significantly, which decreased by 57.32% and 66.67%, respectively (25–31 °C: *t* = 53.43, *df* = 4, *p* < 0.001; 25–34 °C: *t* = 46.30, *df* = 4, *p* < 0.001; Figure 3).

### 3.3. Antioxidant Responses

The standard curves of the antioxidant enzyme activity level detection are shown in Appendix A. The R^2^ values of all antioxidant enzymes are greater than 0.99, indicating that this experiment is real and stable and confirms the validity of the enzyme activity experiment results.

For the SOD and GST enzyme activity levels in the body of *M. hieroglyphica* adults, there was no significant difference with the extension of treatment time at 25, 28, and 31 °C (Appendix A). However, as the temperature increased to 34 °C, the SOD content increased, and the GST content decreased over time (Tukey’s test: SOD: *F*_2, 6_ = 19.40, *p* = 0.002; GST: *F*_2, 6_ = 137.60, *p* < 0.001; Appendix A). The CAT activity levels showed no significant differences with the extension of treatment time at 25 and 28 °C, but the CAT activity levels increased with the extension of treatment time at 31 °C and 34 °C (Tukey’s test: 31 °C: *F*_2, 6_ = 20.50, *p* = 0.002; 34 °C: *F*_2, 6_ = 15.57, *p* = 0.004; Appendix A). The POD activity levels were more sensitive. With the extension of treatment time, it showed a decreasing trend at 25 and 34 °C (Tukey’s test: 25 °C: *F*_2, 6_ = 28.36, *p* = 0.001; 34 °C: *F*_2, 6_ = 126.60, *p* < 0.001; Appendix A), a “first decreasing and then increasing” trend at 28 °C (Tukey’s test: *F*_2, 6_ = 8.49, *p* = 0.018, Appendix A), and an increasing trend at 31 °C (Tukey’s test: *F*_2, 6_ = 6.49, *p* = 0.032, Appendix A). Meanwhile, the results of the two-way analysis of variance showed that different temperature treatments showed significant differences (SOD: *F* = 51.14, *p* = 0.000; CAT: *F* = 38.61, *p* = 0.000; GST: *F* = 38.19, *p* = 0.000; POD: *F* = 11.62, *p* = 0.000; Appendix A), indicating that the main effect of the temperature existed, and different temperatures would have a different relationship with the enzyme activity in *M. hieroglyphica* adults. There was no significant difference between the different treatment times (SOD: *F* = 1.48, *p* = 0.301; CAT: *F* = 2.44, *p* = 0.168; GST: *F* = 1.18, *p* = 0.369; POD: *F* = 0.78, *p* = 0.501; Appendix A), indicating that the main effect of the treatment time did not exist, and different treatment times could not have different relationships with the enzyme activity in *M. hieroglyphica* adults.

Under the same treatment time, the SOD and CAT activity levels increased with the increase in treatment temperature (Tukey’s test: SOD: 24 h: *F*_3, 8_ = 31.20, *p* < 0.001; 48 h: *F*_3, 8_ = 20.96, *p* < 0.001; 72 h: *F*_3, 8_ = 115.33, *p* < 0.001; CAT: 24 h: *F*_3,8_ = 59.93, *p* < 0.001; 48 h: *F*_3, 8_ = 89.76, *p* < 0.001; 72 h: *F*_3, 8_ = 147.73, *p* < 0.001; Figure 4A,B). The GST activity levels showed a decreasing trend (Tukey’s test: 24 h: *F*_3,8_ = 256.72, *p* < 0.001; 48 h: *F*_3, 8_ = 563.66, *p* < 0.001; 72 h: *F*_3, 8_ = 528.40, *p* < 0.001; Figure 4C). The POD activity levels showed a trend of “first decreasing and then increasing” (Tukey’s test: 24 h: *F*_3, 8_ = 83.66, *p* < 0.001; 48 h: *F*_3, 8_ = 254.16, *p* < 0.001; 72 h: *F*_3, 8_ = 217.10, *p* < 0.001; Figure 4D). The above indicates that different antioxidant enzymes have different sensitivity to high temperatures.

## 4. Discussion

Temperatures affect various life histories and behavioral and physiological parameters of insects such as leaf beetles [34] and thereby impact predictions, forecasting, and effective control to varying (and often unpredictable) extents. The adult longevity of *M. hieroglyphica*, from the current study, falls within the lower range when compared to other studies [16]. Such differences might depend on factors that include the diet, adaptation of species to certain climatic conditions, and genetic backgrounds [35,36,37]. Meanwhile, our results also showed that the fecundity of the leaf beetle (*M. hieroglyphica*) significantly decreased with an increasing temperature (25 °C to 32 °C), and it could not lay eggs at 34 °C. This trend is consistent with the study by Wang et al. [38], where the temperature increases from 24 °C to 32 °C, and the fecundity of *Phaedon brassicae* Baly adults gradually decreases until it disappears. Similar findings from Stewart et al. [39] were reported, with *Agasicles hygrophila* Selman & Vogt having the largest spawning at 25 °C and small spawning at 30 °C. Our study findings are similar to those of previous findings, and the fecundity of adult *M. hieroglyphica* is the highest at 25 °C. At the same time, we also found that the daily fecundity of *M. hieroglyphica* remained at 0 for a period of time (15–29 days) during the entire experiment. We speculate that temperature may affect the reproductive pattern of the leaf beetles (*M. hieroglyphica*) throughout its entire period, but the specific reasons need further research. The respective physiological age-related maternity curves indicate differing levels of reproductive potential for the studied populations of *M. hieroglyphica* adults, but they fall within the range of realistic levels when compared to previous studies [16]. Ambient temperatures are only estimates of the temperatures experienced by this insect. The adult *M. hieroglyphica* is highly mobile and can migrate to more optimal temperatures, impacting the final outcome. Similarly, the insect can manipulate its position in the tree canopy to take advantage of the cooling effects from leaf transpiration and shade [16,40]. This aspect is, however, outside the scope of the current study.

Temperature is an important environmental factor affecting insect feeding, and there are differences in insect food intake at different temperatures. The maximum food intake is achieved at the optimal temperature, and the food intake will decrease if it is higher (or lower) than the optimal temperature, thereby affecting the growth and development of insects [41,42]. The results of the influence of temperature on the food intake of adults *Brontispa longissima* (Gestro) showed that the maximum food intake of adults was at 29 °C, and the food intake decreased when the temperature exceeded 29 °C [43]. At the same time, there were similar results in this study; the optimal feeding temperature of adults was between 25 and 28 °C, and the food intake decreased significantly after the temperature increased (31 and 34 °C). Therefore, it is speculated that a high temperature may inhibit certain functions of insects and thus reduce their food intake; however, which specific internal functions [44,45,46] of insects that are affected need further research.

In the present study, we explored the effects of different temperatures on enzymatic antioxidant defense systems (SOD, CAT, GST, and POD) of the leaf beetle *M. hieroglyphica*. Our results showed that in the first 3 days, when the temperature increased to 34 °C, the activities of SOD and CAT gradually increased with the treatment time, while the activities of GST and POD gradually decreased. SOD and CAT enzymes are important enzymes in the antioxidant response system, which play a role in scavenging high-concentration superoxide anion radicals and decomposing high-concentration H_2_O_2_ to protect insects from oxidative damage [47,48]. Some scholars have confirmed that SOD and CAT enzymes are the first and most important defense against ROS, and the level of their activity directly or indirectly determines the ability of insects to resist thermal stress [49,50]. Previous studies have shown that SOD and CAT enzyme activities increase in insects at high temperatures, for example, *Propylaea japonica* Thunberg [30], *Chilo suppressalis* Walker [51], *Bactrocera dorsalis* Hendel [52], and *Ophraella communa* LeSage [53]. The above results are consistent with the results of this study, so we conclude that SOD and CAT enzymes have a protective effect on *M. hieroglyphica* under thermal stress (34 °C).

## 5. Conclusions

In this study, we investigated the effects of different temperatures (25, 28, 31, and 34 °C) on the survival, reproduction, feeding capacity, and antioxidant capacity of adult leaf beetles. Our findings indicate that high temperatures (31 and 34 °C) have a negative impact on the survival and reproduction of *M. hieroglyphica* adults. The temperature-mediated effects are also reflected in the feeding capacity of *M. hieroglyphica* adults: the feeding capacity of adults was strongest at 25 °C, while the feeding capacity of adults steadily decreased at higher temperatures. The above life history and feeding capacity impacts are also reflected in the antioxidant capacity of leaf beetle adults, with SOD, CAT, GST, and POD all affected by temperature. Thus, laboratory assays can help to understand the response of *M. hieroglyphica* adults to the temperature from ecological and physiological research perspectives.

## Figures and Tables

**Figure 1 insects-16-00222-f001:**
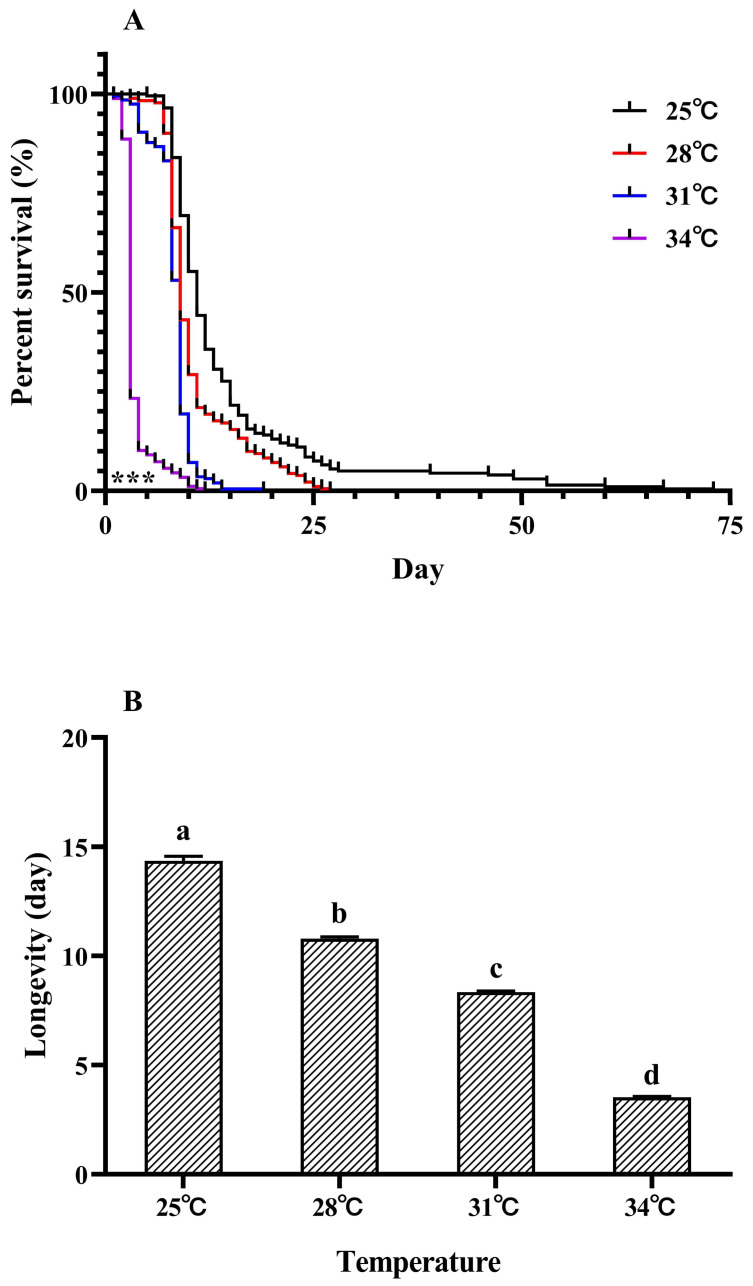
Survival curves of adult *Monolepta hieroglyphica* (**A**) at different temperatures. (**B**) Adult longevity of *Monolepta hieroglyphica*. Survival statistics were calculated using the Kaplan–Meier survival curve and compared using the log-rank test (individuals = 300, *** *p* < 0.001). For each species, different letters above the bars indicate statistically significant differences (ANOVA: Tukey’s post hoc test, *p* < 0.05).

**Figure 2 insects-16-00222-f002:**
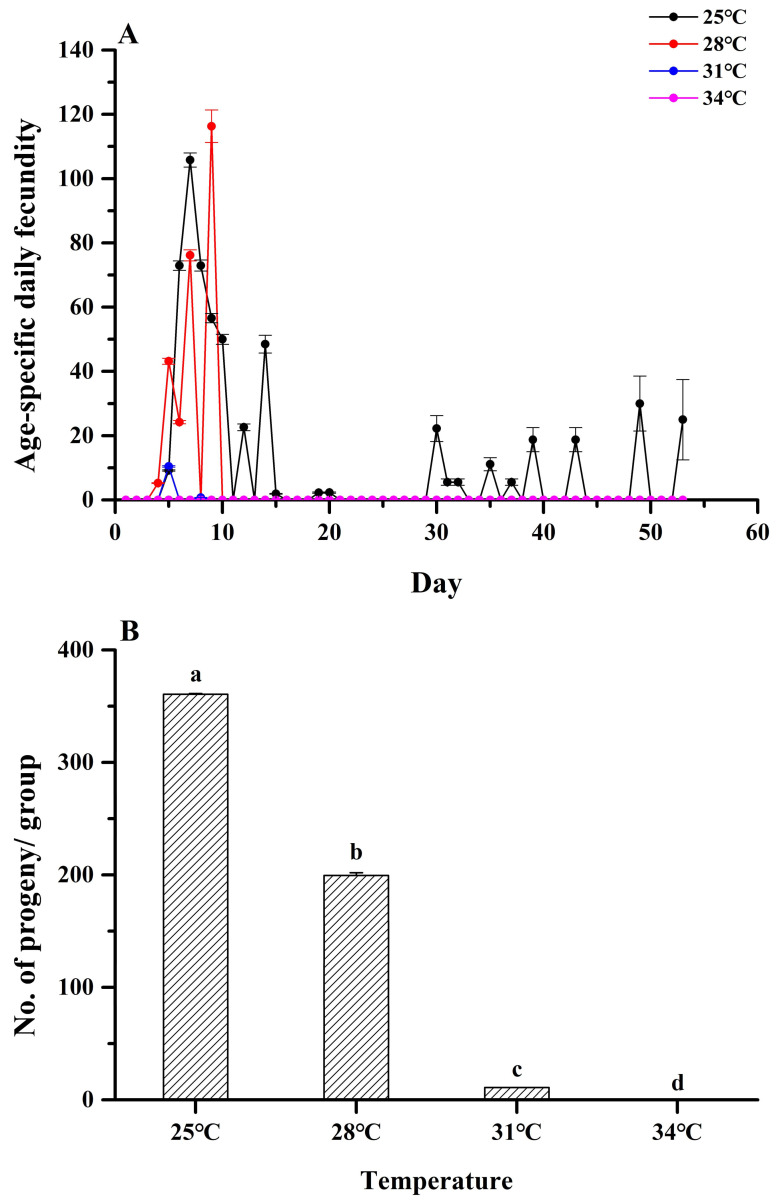
Age-specific fecundity of adult *Monolepta hieroglyphica* (**A**) at different temperatures. (**B**) Fecundity of *Monolepta hieroglyphica*. For each species, different letters above the bars indicate statistically significant differences among temperatures (ANOVA: Tukey’s post hoc test, *p* < 0.05).

**Figure 3 insects-16-00222-f003:**
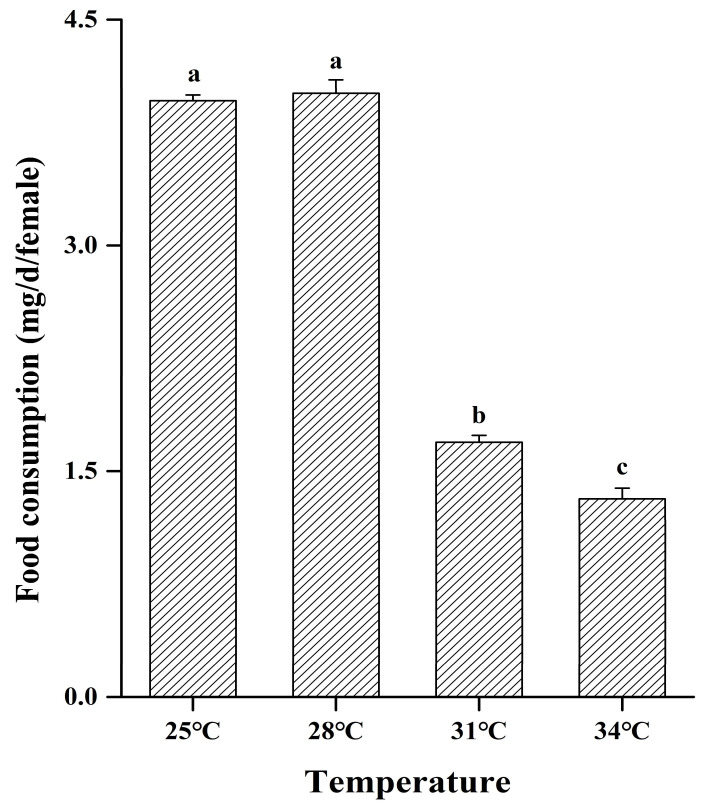
Daily food consumption of adult *Monolepta hieroglyphica* at different temperatures. Different letters above the bars indicate statistically significant differences among temperatures (ANOVA: Tukey’s post hoc test, *p* < 0.05).

**Figure 4 insects-16-00222-f004:**
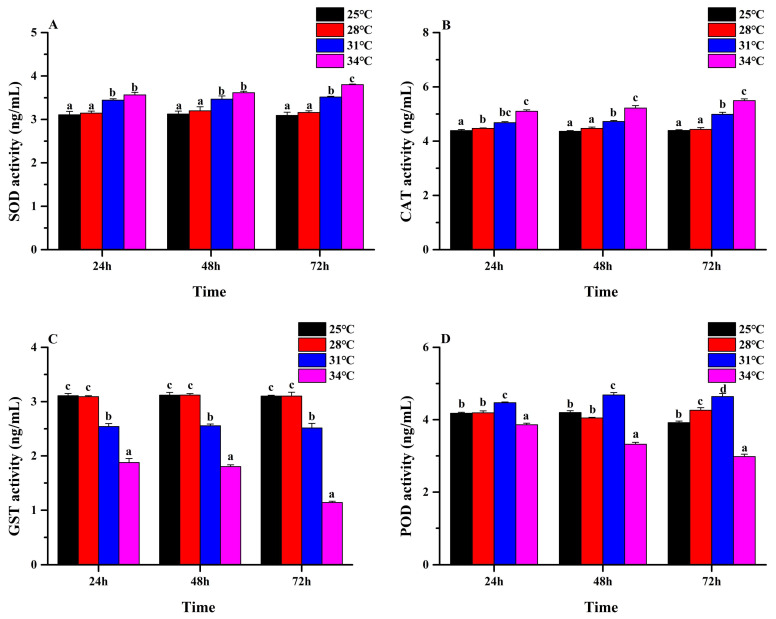
Effects of temperature stress on the detoxification enzyme activity levels of adult *Monolepta hieroglyphica*. (**A**): SOD activity; (**B**): CAT activity; (**C**): GST activity; (**D**): POD activity. Different letters above the bars indicate statistically significant differences at *p* < 0.05 (ANOVA followed by a Tukey’s post hoc test).

## Data Availability

All data analyzed in this study are included in this article.

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
