# Peer review of "Impact of Different Temperatures on Activity of the Pest *Monolepta hieroglyphica* Motschulsky (Coleoptera: Chrysomelidae)"

_insects, 2025, doi:10.3390/insects16020222_

Round 1
Reviewer 1 Report
Comments and Suggestions for Authors
In this study, the authors investigated the effects of different temperatures on Monolepta hieroglyphica, including the survival, longevity, fecundity, feeding capacity, and antioxidant capacity. The methods used in this manuscript are scientifically sound, and the results are also clearly presented. However, there were some questions listed as follows:
1. Why did the authors select these four temperatures for the experiments? After all, the temperature set by the author is not too high.
2. For the experiments of adult feeding capacity, why did the authors only select the females for the experiments? Maybe the results of males also should be added.
3. For the results of figure 2A, we found that essentially no fecundity was observed on days 15-29 at 25 ℃. It is an interesting phenomenon. The authors could have explained it in discussion.
4. The figure 4 should be removed as a supplemental figure.
5. The results of Table 1 and Figure 5 presented in this manuscript were the same results. Therefore, the authors need to remove one of them. In addition, the letters in table 1 should be annotated.
6. For figure 5C, the letters for 48 and 72 h were wrong.
7. Please check the manuscript more carefully. Some format or syntax errors can be found. For example, line 52, Monolepta hieroglyphica should be M. hieroglyphica.
Author Response
Reviewer #1
Open Review ( ) I would not like to sign my review report
(x) I would like to sign my review report
Quality of English Language
(x) The quality of English does not limit my understanding of the research.
( ) The English could be improved to more clearly express the research.
|
Yes |
Can be improved |
Must be improved |
Not applicable |
|
|
Does the introduction provide sufficient background and include all relevant references? |
( ) |
(x) |
( ) |
( ) |
|
Is the research design appropriate? |
( ) |
(x) |
( ) |
( ) |
|
Are the methods adequately described? |
( ) |
(x) |
( ) |
( ) |
|
Are the results clearly presented? |
( ) |
(x) |
( ) |
( ) |
|
Are the conclusions supported by the results? |
( ) |
(x) |
( ) |
( ) |
Response: Thank you very much for your comments. We have revised the manuscript as suggested
Comments and Suggestions for Authors
In this study, the authors investigated the effects of different temperatures on Monolepta hieroglyphica, including the survival, longevity, fecundity, feeding capacity, and antioxidant capacity. The methods used in this manuscript are scientifically sound, and the results are also clearly presented. However, there were some questions listed as follows:
Q1: Why did the authors select these four temperatures for the experiments? After all, the temperature set by the author is not too high.
Response: We have added and revised the manuscript as suggested. (Lines 107-111)
Lines 107-111: All subsequent experimental assays were performed in the lab and carried out at climatic chambers with the temperatures of 25 ℃, 28 ℃, 31 ℃, and 34 ℃, which mirror the prevailing temperatures during the corn and other crops growing season under the background of greenhouse effect in north China. Combined with the pre-experimental results, the above four temperatures (25 ℃, 28 ℃, 31 ℃, and 34 ℃) were determined.
Q2: For the experiments of adult feeding capacity, why did the authors only select the females for the experiments? Maybe the results of males also should be added.
Response: Thank you very much for your comments. We only monitored the feeding capacity of female adults, and more systematic studies will be conducted to increase monitoring of male adults.
Q3: For the results of figure 2A, we found that essentially no fecundity was observed on days 15-29 at 25 ℃. It is an interesting phenomenon. The authors could have explained it in discussion.
Response: Thanks. We have revised the manuscript as suggested (Lines 281-285).
Lines 281-285: At the same time, we also found that the daily fecundity of M. hieroglyphica remained at 0 for a period of time (15-29 days) during the entire experiment. We speculate that temperature may affect the reproductive pattern of the leaf beetles (M. hieroglyphica) throughout its entire period, but the specific reasons need further research.
Q4: The figure 4 should be removed as a supplemental figure.
Response: Thanks. We have revised the manuscript as suggested, and Figure 4 has been changed to Supplemental Figure S1.
Q5: The results of Table 1 and Figure 5 presented in this manuscript were the same results. Therefore, the authors need to remove one of them. In addition, the letters in table 1 should be annotated.
Response: Thanks. We have revised the manuscript as suggested. The Table 1 was deleted and was modified to Supplemental Table S1in this manuscript, and the letters in Supplemental Table S1 was annotated.
Q6: For figure 5C, the letters for 48 and 72 h were wrong.
Response: We have now revised the Figure as suggested.
Q7: Please check the manuscript more carefully. Some format or syntax errors can be found. For example, line 52, Monolepta hieroglyphica should be M. hieroglyphica.
Response: Response: Thanks. We have checked and revised the manuscript as suggested.
Reviewer 2 Report
Comments and Suggestions for Authors
Specific comments and suggestions:
Simple Summary & Abstract:
L.13 & 15: Delete “i.e”
L.19: Replace the “thermoal biology” with “thermal biology”
L.19: Delete the “individual” before “leaf beetles”
L.23: Delete “etc” and add “and” before “millet”
L.23-25: Replace this “In recent years, the greenhouse effect has been severe, so we wanted to understand the adaptability of M. hieroglyphica adults to temperature.” with “Given the increasing severity of the greenhouse effect in recent years, we aimed to investigate the adaptability of M. hieroglyphica adults to varying temperatures." to improve clarity.
L.28-29: Change “Similarly, temperature has a negative effect on the feeding capacity of M. hieroglyphica adults, and the negative effect was greater with the increase of temperature.” to "Similarly, temperature negatively affected the feeding capacity of M. hieroglyphica adults, with the impact becoming more pronounced as the temperature increased."
L.32: “The POD activity levels showed a trend of first decreasing and then increasing” can be modified as “The POD activity showed a biphasic response to increasing temperatures, first decreasing and then increasing.”
L33-33: Reword these lines to improve clarity.
Introduction:
L.56: Be consistent when using punctuation marks (e.g. comma), especially when listing more than two items (thus, “.....growth, development, and survival.......”). Make corrections throughout the text.
L.61-62: Replace this “for Monolepta hieroglyph (Motschulsky adults” with “....for M. hieroglyphica adults”
L.65: Delete “(temporary)”
L.59-67: Based on lines 59 to 62 what makes the current research (lines 64 to 67) relevant/novel? Can the authors add more relevant literature to justify this in the text?
L.74: Spell out “H2O2”; this can be put in parenthesis/bracket. Make such corrections when using chemical formulas for the first time through the text.
Materials and Methods:
L.93: Is it the corn plant or part of the plant? If a whole plant was used, specify the stage/age of the corn plant used.
L.97: Replace “stabilized” with “stability”. Also, delete “L” after Zea mays.
L.98: Delete “and” after “population; reword as “The F1 progeny was used for this experiment”
L.96-98: Why was 25 ̊C chosen for culturing the leaf beetle? What stage/age of the F1 progeny was used? Add to the text
L.114: Change to “ All climatic chambers were monitored using HOBO to ensure constant temperatures were maintained”
L.116: “.... a corn”; corn plant or part of corn plant? Specify
L.117: Check the “Observations”
L.118: Replace “fifty pairs” with “50 pairs”
L.121: Replace “According to Kaufmann” with “As described by Kaufmann”
L128-129: Change “number of food” to “quantity of food”. Also, “....we recorded the quantity of corn......”
L.140: “...healthy adults...”; Were they sexed (male or female)? If not, could this affect the activity levels of any of the four antioxidant enzymes?
Results:
L.166: Reword these lines: 166-169
L.181: Lines 181 to 185 are not clear; “Peak oviposition..........of M. hieroglyphica.” Reword it.
L.187: Figure 2A, was the observed reduction in the daily fecundity positively associated with the increased mortality of the adult females over the experimental duration?
L.208-230: Table 1 data analysis and interpretation were not clear; the use of upper and lowercase alphabets were not explained in the table’s title or as a footnote; two-way ANOVA (Time and Temperature) could have been used to ensure better interpretation of the table’s results. Also, in Fig. 5, instead of the one-way ANOVA, the authors can use two-way ANOVA (Time and Temperature) to better explain each enzyme's activity trend.
Discussion and Conclusions:
L.244 and 246: In-text citation should be avoided throughout the text
L.248: “we also infer that 25 ℃ is the most suitable egg-laying temperature for M. hieroglyphica” How did the rearing condition influence the oviposition of leaf beetles at different temperatures? Can we deduce that the leaf beetles were already acclimated to 25 ℃ which resulted in the highest number of eggs at said temperature?
L265: Delete “(no significant difference)”

There is room for additional edits to enhance readability and soundness.
Author Response
Reviewer #2
Open Review
( ) I would not like to sign my review report
(x) I would like to sign my review report
Quality of English Language
( ) The English is fine and does not require any improvement.
(x) The English could be improved to more clearly express the research.
|
Yes |
Can be improved |
Must be improved |
Not applicable |
|
|
Does the introduction provide sufficient background and include all relevant references? |
( ) |
(x) |
( ) |
( ) |
|
Is the research design appropriate? |
( ) |
(x) |
( ) |
( ) |
|
Are the methods adequately described? |
( ) |
( ) |
(x) |
( ) |
|
Are the results clearly presented? |
( ) |
( ) |
(x) |
( ) |
|
Are the conclusions supported by the results? |
( ) |
( ) |
(x) |
( ) |
Response: Thank you very much for your comments. The manuscript has been edited by native English speakers.
Comments and Suggestions for Authors
Specific comments and suggestions:
Simple Summary & Abstract:
Q1-Q8: L.13 & 15: Delete “i.e”
L.19: Replace the “thermoal biology” with “thermal biology”
L.19: Delete the “individual” before “leaf beetles”
L.23: Delete “etc” and add “and” before “millet”
L.23-25: Replace this “In recent years, the greenhouse effect has been severe, so we wanted to understand the adaptability of M. hieroglyphica adults to temperature.” with “Given the increasing severity of the greenhouse effect in recent years, we aimed to investigate the adaptability of M. hieroglyphica adults to varying temperatures." to improve clarity.
L.28-29: Change “Similarly, temperature has a negative effect on the feeding capacity of M. hieroglyphica adults, and the negative effect was greater with the increase of temperature.” to "Similarly, temperature negatively affected the feeding capacity of M. hieroglyphica adults, with the impact becoming more pronounced as the temperature increased."
L.32: “The POD activity levels showed a trend of first decreasing and then increasing” can be modified as “The POD activity showed a biphasic response to increasing temperatures, first decreasing and then increasing.”
L33-33: Reword these lines to improve clarity.
Response: Thank you very much for your comments. We have revised the manuscript as suggested.
Introduction:
Q9-Q13: L.56: Be consistent when using punctuation marks (e.g. comma), especially when listing more than two items (thus, “.....growth, development, and survival.......”). Make corrections throughout the text.
L.61-62: Replace this “for Monolepta hieroglyph (Motschulsky adults” with “....for M. hieroglyphica adults”
L.65: Delete “(temporary)”
L.59-67: Based on lines 59 to 62 what makes the current research (lines 64 to 67) relevant/novel? Can the authors add more relevant literature to justify this in the text?
L.74: Spell out “H2O2”; this can be put in parenthesis/bracket. Make such corrections when using chemical formulas for the first time through the text.
Response: Thank you very much for your comments. We have revised the manuscript as suggested.
Materials and Methods:
Q14-Q24: L.93: Is it the corn plant or part of the plant? If a whole plant was used, specify the stage/age of the corn plant used.
L.97: Replace “stabilized” with “stability”. Also, delete “L” after Zea mays.
L.98: Delete “and” after “population; reword as “The F1 progeny was used for this experiment”
L.96-98: Why was 25 ̊C chosen for culturing the leaf beetle? What stage/age of the F1 progeny was used? Add to the text
L.114: Change to “ All climatic chambers were monitored using HOBO to ensure constant temperatures were maintained”
L.116: “.... a corn”; corn plant or part of corn plant? Specify
L.117: Check the “Observations”
L.118: Replace “fifty pairs” with “50 pairs”
L.121: Replace “According to Kaufmann” with “As described by Kaufmann”
L128-129: Change “number of food” to “quantity of food”. Also, “....we recorded the quantity of corn......”
L.140: “...healthy adults...”; Were they sexed (male or female)? If not, could this affect the activity levels of any of the four antioxidant enzymes?
Response: Thank you very much for your comments. We have revised the manuscript as suggested.
eg:
Lines 104-110: M. hieroglyphica was reared on Zea mays L. (corn: five-leaf stage of variety Longsheng 802, Jinzhong Longsheng Seed Co., Ltd., Shanxi, China) in screened cages (30 × 30 × 30 cm) within a controlled climate chamber (RXZ500D, Ningbo Jiangnan Instrument Factory, Ningbo, China) and held at 25 ± 1 ℃, 70 ± 5% RH, and 16:8 h (Light: Dark) photoperiod. Zea mays was replaced every two days to ensure the stability of the M. hieroglyphica population. The F1 progeny was used for this experiment.
Lines 118-119: F1 adults of M. hieroglyphica < 12 h of age were removed from the rearing colony, placed within screened cages (30 × 30 × 30 cm) and fed corn ad libitum.
Lines129-130: Each 50 pair of leaf beetles were placed in screened cages (30 × 30 × 30 cm), containing a egg box and a corn (five-leaf stage of corn plant).
Lines 155: Next, live, healthy adults (male and female) were chosen
Results:
Q25-Q28: L.166: Reword these lines: 166-169
L.181: Lines 181 to 185 are not clear; “Peak oviposition..........of M. hieroglyphica.” Reword it.
L.187: Figure 2A, was the observed reduction in the daily fecundity positively associated with the increased mortality of the adult females over the experimental duration?
L.208-230: Table 1 data analysis and interpretation were not clear; the use of upper and lowercase alphabets were not explained in the table’s title or as a footnote; two-way ANOVA (Time and Temperature) could have been used to ensure better interpretation of the table’s results. Also, in Fig. 5, instead of the one-way ANOVA, the authors can use two-way ANOVA (Time and Temperature) to better explain each enzyme's activity trend.
Response: Thank you very much for your comments. We have revised the manuscript as suggested.
eg:
Lines183-186: At 34℃, the survival rate of M. hieroglyphica adults decreased most significantly (Tukey's test: F3,8 = 6040.50, p < 0.001, Figure 1A). There was a significant difference in the survival rate of M. hieroglyphica adults on the first day of treatment compared to other temperatures
Lines 207: Figure 2A. We think there is an impact, but it's not significant. Because survival rate results at 25,28, and 31 ℃ showed no significant difference in the first five days, but fecundity results showed significant differences.
L.208-230: We have revised and added the manuscript as suggested.
Discussion and Conclusions:
Q29 and Q31: L.244 and 246: In-text citation should be avoided throughout the text
L.248: “we also infer that 25 ℃ is the most suitable egg-laying temperature for M. hieroglyphica” How did the rearing condition influence the oviposition of leaf beetles at different temperatures? Can we deduce that the leaf beetles were already acclimated to 25 ℃ which resulted in the highest number of eggs at said temperature?
L265: Delete “(no significant difference)”
Response: Thank you very much for your comments. We have revised the manuscript as suggested.
eg:
Lines 279-280: Our study findings are similar to those of previous findings, and the fecundity of adult M. hieroglyphica is the highest at 25℃
Reviewer 3 Report
Comments and Suggestions for Authors
Dear authors,
Your manuscript is quite good in terms of subject matter, which is obviously of interest.
However, some improvement suggestions are needed tofor clarification and increasing the value of your work.
Lines 2-3: I would change the title because it seems incomplete in its current form, i.e.: <Impact of different temperatures on Monolepta hieroglyphica Motschulsky (Coleoptera: Chrysomelidae)
Why? Because it lacks the process in the insect's life on which the temperature factor will or will not have an effect. So I suggest the following: Impact of different temperatures on activity of the pest Monolepta hieroglyphica Motschulsky (Coleoptera: Chrysomelidae).>
Line 37: Having carefully analyzed the manuscript and the results, I would recommend removing the keyword <IPM> because it has little to do with the topic addressed and is not supported by the results.
Line 46: Related to: <This species was capable of shifting its primary threat among various cash crops and weedscrops and weeds>, I suggest detailing list of crops and weeds described in these sources (4,5) so that the reader can understand better, without having to search.
Line 51: Related to: <Furthermore, the geographical range of M. hieroglyphica has expanded, and its rate of spread has accelerated [6], marking this insect as a key concern.>
Please mention a few words about the distribution area, as it is a pest localized to Asia from what I have found in the literature. It is essential to point out the areas with large pest populations in order to see the overall picture.
Lines 54-85: Related to: <The data obtained will provide a theoretical basis for predicting and forecasting, as well as for the effective control of this leaf beetle. >
Either develop or remove the statement because it is not justified, it is just an unfounded assumption.
Line 89: Related to: <Key characteristics were used to determine the species of M. hieroglyphica >
What key features are you referring to? Please elaborate.
Line 96: What is L:D? Please elaborate in parentheses.
Lines 100-101: Related to: <All subsequent experimental assays were performed in the lab and carried out at climatic chambers with the temperatures of 25 ℃, 28 ℃, 31 ℃, and 34 ℃,...>.
Were these temperature values chosen randomly or according to the literature? Please mention, somewhere in parentheses, the method of choice.
Line 130: Does the formula belong to you or is it consecrated? Ensure its ownership by mentioning the Formula of...
Lines 46-147: Related to: <There were three replicates for each treatment and one pair of leaf beetles in each replicate. >
In Material and Method you talk about pairs of beetles, meaning females and males I assume. Is there any difference between females and males in the evaluation? Or did you quantify them in bulk? Because in Results I did not find any analysis/interpretation by sex. That is why I do not see the logic of choosing in pairs.
Lines 204-206: Related to: <The standard curves of antioxidant enzyme activity levels detection were shown in Figure 4. The R2 values of all antioxidant>.
The interpretation of Figure 4 is superficial. Expand on the existing text and compare the enzymatic activities.
Lines 296-297: .. <.and inform strategies to integrated pest management (IPM) under conditions of global warming or extreme weather events. >
The results and data presented in the manuscript do not suggest any connection with management strategies such as IPM, So I suggest you remove this sentence because it is not sufficiently supported in the Results chapter.
Line 310: Please, standardize the description of References. For example, regarding the titles of journals, in the first part (up to Ref#22) they are mentioned in full, then some abbreviations appear (e.g. Ref#23, Ref#25, Ref#27...etc).
Kind regards,
R

Author Response
Reviewer #3
Open Review ( ) I would not like to sign my review report
(x) I would like to sign my review report
Quality of English Language
(x) The English is fine and does not require any improvement.
( ) The English could be improved to more clearly express the research.
|
Yes |
Can be improved |
Must be improved |
Not applicable |
|
|
Does the introduction provide sufficient background and include all relevant references? |
( ) |
(x) |
( ) |
( ) |
|
Is the research design appropriate? |
( ) |
(x) |
( ) |
( ) |
|
Are the methods adequately described? |
( ) |
(x) |
( ) |
( ) |
|
Are the results clearly presented? |
( ) |
(x) |
( ) |
( ) |
|
Are the conclusions supported by the results? |
(x) |
( ) |
( ) |
( ) |
Response: Thank you very much for your comments. We have revised the manuscript as suggested.
Comments and Suggestions for Authors:
Dear authors,
Your manuscript is quite good in terms of subject matter, which is obviously of interest.
However, some improvement suggestions are needed tofor clarification and increasing the value of your work.
Q1: Lines 2-3: I would change the title because it seems incomplete in its current form, i.e.: <Impact of different temperatures on Monolepta hieroglyphica Motschulsky (Coleoptera: Chrysomelidae)
Why? Because it lacks the process in the insect's life on which the temperature factor will or will not have an effect. So I suggest the following: Impact of different temperatures on activity of the pest Monolepta hieroglyphica Motschulsky (Coleoptera: Chrysomelidae).
Response: Thank you very much for your comments. We have revised the title of manuscript as suggested (Lines 2-3)
Lines 2-3: Impact of Different Temperatures on activity of the pest Monolepta hieroglyphica Motschulsky (Coleoptera: Chrysomelidae)
Q2: Line 37: Having carefully analyzed the manuscript and the results, I would recommend removing the keyword <IPM> because it has little to do with the topic addressed and is not supported by the results.
Response: Thank, we have revised the manuscript as suggested.
Q3: Line 46: Related to: <This species was capable of shifting its primary threat among various cash crops and weedscrops and weeds>, I suggest detailing list of crops and weeds described in these sources (4,5) so that the reader can understand better, without having to search.
Response: We have revised the manuscript as suggested (Lines 51-53).
Lines 51-53: This species was capable of shifting its primary threat among various cash crops and weeds, such as corn, cotton, purslane, etc [4,5].
Q4: Line 51: Related to: <Furthermore, the geographical range of M. hieroglyphica has expanded, and its rate of spread has accelerated [6], marking this insect as a key concern.>
Please mention a few words about the distribution area, as it is a pest localized to Asia from what I have found in the literature. It is essential to point out the areas with large pest populations in order to see the overall picture.
Response: We have revised the manuscript as suggested (Lines 56-60).
Lines 56-60: Furthermore, the geographical range of M. hieroglyphica has expanded, starting from Heilongjiang and Inner Mongolia in the north, extending to Taiwan, Guangdong, Guangxi, and Yunnan in the south, reaching the border line in the east, and extending to Ningxia, Gansu, Sichuan, and Yunnan in the west. Its rate of spread has accelerated [6], marking this insect as a key concern.
Q5: Lines 54-85: Related to: <The data obtained will provide a theoretical basis for predicting and forecasting, as well as for the effective control of this leaf beetle. >
Either develop or remove the statement because it is not justified, it is just an unfounded assumption.
Response: We have revised the manuscript as suggested (Lines 92-94).
Lines 92-94: The data obtained will provide a theoretical basis for the development of integrated pest management (IPM) programs of this leaf beetle.
Q6: Line 89: Related to: <Key characteristics were used to determine the species of M. hieroglyphica >
What key features are you referring to? Please elaborate.
Response: We have revised the manuscript as suggested (Lines 99-101).
Lines 99-101: Key characteristics were used to determine the species of M. hieroglyphica, for example, a near-circular pale spot was found on each elytra base, etc [31,32].
Q7: Line 96: What is L:D? Please elaborate in parentheses.
Response: We have revised the manuscript as suggested (Lines 107-108).
Q8: Lines 107-108: 16:8 h (Light: Dark) photoperiod.
Lines 100-101: Related to: <All subsequent experimental assays were performed in the lab and carried out at climatic chambers with the temperatures of 25 ℃, 28 ℃, 31 ℃, and 34 ℃,...>.
Were these temperature values chosen randomly or according to the literature? Please mention, somewhere in parentheses, the method of choice.
Response: We have revised the manuscript as suggested (Lines 112-116).
Lines 112-116: All subsequent experimental assays were performed in the lab and carried out at climatic chambers with the temperatures of 25 ℃, 28 ℃, 31 ℃, and 34 ℃, which mirror the prevailing temperatures during the corn and other crops growing season under the background of greenhouse effect in north China. Combined with the pre-experimental results, the above four temperatures (25 ℃, 28 ℃, 31 ℃, and 34 ℃) were determined.
Q9: Line 130: Does the formula belong to you or is it consecrated? Ensure its ownership by mentioning the Formula of...
Response: We have revised the manuscript as suggested and added the references after the formula.
Q10: Lines 146-147: Related to: <There were three replicates for each treatment and one pair of leaf beetles in each replicate. >
In Material and Method you talk about pairs of beetles, meaning females and males I assume. Is there any difference between females and males in the evaluation? Or did you quantify them in bulk? Because in Results I did not find any analysis/interpretation by sex. That is why I do not see the logic of choosing in pairs.
Response: Thank you very much for your comments. Two leaf beetles were used for each replicate in this experiment. To avoid gender differences (no experiment was conducted to prove whether there were differences between genders), we standardized each replicate (one pair of leaf beetles).
Q11: Lines 204-206: Related to: <The standard curves of antioxidant enzyme activity levels detection were shown in Figure 4. The R2 values of all antioxidant>.
The interpretation of Figure 4 is superficial. Expand on the existing text and compare the enzymatic activities.
Response: We have revised the manuscript as suggested (Lines 219-222).
Lines 219-222: The standard curves of antioxidant enzyme activity levels detection were shown in Supplemental Figure S1. The R2 values of all antioxidant enzymes are greater than 0.99, indicating that this experiment is real and stable. And confirm the validity of the enzyme activity experiment results.
Q12: Lines 296-297: .. <.and inform strategies to integrated pest management (IPM) under conditions of global warming or extreme weather events. >
The results and data presented in the manuscript do not suggest any connection with management strategies such as IPM, So I suggest you remove this sentence because it is not sufficiently supported in the Results chapter.
Response: We have revised the manuscript as suggested.
Q13: Line 310: Please, standardize the description of References. For example, regarding the titles of journals, in the first part (up to Ref#22) they are mentioned in full, then some abbreviations appear (e.g. Ref#23, Ref#25, Ref#27...etc).
Response: We have revised the references in manuscript as suggested.
Round 2
Reviewer 3 Report
Comments and Suggestions for Authors
The comments and suggestions for improvement have been successfully resolved.
Author Response
Open Review: ( ) I would not like to sign my review report
(x) I would like to sign my review report
Quality of English Language
(x) The English is fine and does not require any improvement.
( ) The English could be improved to more clearly express the research..
|
Yes |
Can be improved |
Must be improved |
Not applicable |
|
|
Does the introduction provide sufficient background and include all relevant references? |
(x) |
( ) |
( ) |
( ) |
|
Is the research design appropriate? |
(x) |
( ) |
( ) |
( ) |
|
Are the methods adequately described? |
(x) |
( ) |
( ) |
( ) |
|
Are the results clearly presented? |
(x) |
( ) |
( ) |
( ) |
|
Are the conclusions supported by the results? |
(x) |
( ) |
( ) |
( ) |
Comments and Suggestions for Authors
The comments and suggestions for improvement have been successfully resolved.
Response: Thank you very much for your comments.